# Gestational syphilis in a tertiary health service in Paraná, Brazil: A case-control study

**Fernando Braz Pauli**[1�‸], **Valdir Spada Júnior**[2], **Renan William Mesquita**[2☸],
**Guilherme Welter Wendt**[1☸], **Paulo Cezar Nunes Fortes**[2], **Harapan Harapan**[3], **Lirane Elize Defante Ferreto**[1☸]*

**1** Postgraduate Program in Applied Health Sciences, Western Paraná State University (UNIOESTE), Francisco Beltrão, Brazil, **2** Health Sciences Center, Western Paraná State University (UNIOESTE), Francisco Beltrão, Brazil, **3** Department of Microbiology, School of Medicine, Universitas Syiah Kuala, Banda Aceh, Indonesia

☸ These authors contributed equally to this work.
* lferreto@gmail.com

**Data Availability Statement:** The underlying data can be found at the following Figshare repository: https://doi.org/10.6084/m9.figshare.25966969.v1.

## Abstract

Approximately 10–12 million new syphilis infections occur annually worldwide, including in pregnant women. This study identified the factors associated with syphilis in pregnant women admitted to a tertiary maternity ward in the State of Paraná, Brazil. This is an ambispective, paired case-control study (1:2 ratio) conducted from September 2020 to October 2021. Pregnant patients (n = 93) admitted to the maternity ward, who were tested with the Venereal Disease Research Laboratory (VDRL) and rapid reagent test, were compared with 186 controls, matched by age and period of hospital admission. Sociodemographic, behavioral, prenatal, and maternity healthcare information was collected through interviews. The data were analyzed using binary logistic regression. Results showed that race/skin color other than white (OR: 2.12; 95%CI: 1.19–3.80; $p < 0.001$), having more than one sexual partner (OR: 3.69; 95%CI: 1.70–8.00; $p = 0.001$), being a former smoker (OR: 2.07; 95%CI: 1.07–4.01; $p = 0.030$) and a current smoker (OR: 4.31; 95%CI: 1.55–11.98; $p = 0.005$), as well as having a history of sexually transmitted infections (OR: 10.87; 95%CI: 4.04–29.27; $p < 0.0.01$) were risk factors for gestational syphilis. In summary, the study indicated that sociodemographic, behavioral, and healthcare-related variables were associated with gestational syphilis. Therefore, practitioners could benefit from incorporating these factors to deliver evidence-based treatment for gestational syphilis.

## Introduction

Around 10 to 12 million new syphilis infections occur each year, and infection rates vary significantly between countries [1]. The infection rate among pregnant women is usually low in high-income countries and is estimated to be around 0.02%–4.5% in Europe and the United States, while it is estimated around 3–18% in Africa [2]. The World Health Organization (WHO) estimates that there are 930,000 pregnant women with active syphilis annually. This has led to approximately 350,000 adverse birth outcomes, including 143,000 early fetal deaths/stillbirths, 62,000 neonatal deaths, 44,000 premature/low birth weight babies, and 102,000 infected babies [3].

**Funding:** Auxílio n˚ 1397/2020 - PROAP/AUXPE – CAPES (Project 88887.594108/2020-00).

**Competing interests:** The authors declare that there are no competing interests.

Given the preventable nature of congenital syphilis, several studies have examined potential missed opportunities to address this situation and highlighted the importance of early and adequate prenatal care, the timely identification of pregnant women with syphilis, and, if infected, the administration of stage-appropriate penicillin at least 30 days before childbirth [4].

In 2016, the PAHO adopted the "Plan of Action for the Prevention and Control of HIV and Sexually Transmitted Infections 2016–2021," setting the goal of accelerating progress towards ending the acquired immunodeficiency syndrome (AIDS) and sexually transmitted infections (STIs) epidemics in the Americas by 2030. However, mother-to-child transmission of congenital syphilis increased during this period, with an incidence rate of 2.3 per thousand live births in 2019, far from the established target of 0.5 cases per thousand live births [1].

Gestational syphilis is an under-recognized problem that burdens the health system and the economy. The babies of pregnant women with syphilis remain in the hospital for longer periods, which inevitably results in higher public health costs. Countries such as Brazil, which have limited resources, could benefit from eliminating the vertical transmission of syphilis [5].

Congenital syphilis is the second most common cause of preventable stillbirth. The period ranging from 2013 to 2018 saw an average annual growth of 25% in the detection rate of pregnant women with syphilis, while the rate increased by 33.8% between 2020 and 2022. It is also observed that although the incidence rate of congenital syphilis has remained stable in this period, it increased by 16% in the comparison between 2022 and 2019, the year before the Covid-19 pandemic [6].

There is a deficiency in the diagnosis of syphilis in pregnant women in the Brazilian public health system. Official data indicate that one third occur in the third trimester, a period in which the repercussions for the fetus are more serious [7]. In addition, according to a study conducted in a maternity hospital in the metropolitan region of Rio de Janeiro, 80% of the cases of congenital syphilis occurred in pregnant women who received prenatal care, but who were not diagnosed or treated until less than 30 days before delivery [8].

Challenges for the Brazilian health system are evident in such a scenario, where there is an alarming epidemic of syphilis in pregnant women. Prematurity, fetal death, and miscarriages, the most well-known consequences of syphilis, combined with follow-up treatment for infected newborns, produce costs for the health system and society that are potentially avoidable [9]. Therefore, data that allow the development of more effective intervention measures for gestational syphilis are needed. Consequently, this study aimed to identify the factors associated with syphilis in pregnant women in Paraná, Brazil.

## Methods

### Study design and setting

A case-control study was conducted between September 2020 and October 2021 in a tertiary-level maternity hospital (Hospital Regional do Sudoeste *Walter Alberto Pecóits*; HRS hospital), located in the state of Paraná, Brazil. The hospital is a referral center in the National Health Service, or SUS. For intermediate-risk and high-risk pregnancies for the 27 municipal districts in the area, covered by the 8th Local Health Authority of the State of Paraná [10].

All pregnant women who were admitted to the maternity ward of the hospital unit with a positive diagnosis of syphilis, coming from any of the 27 municipalities in the area covered by the 8th Health Region, were selected to participate in this investigation. In addition, two controls were included for each case, representing pregnant women who were hospitalized in the same period but did not show reactivity for syphilis in their exams.

To define the cases, we considered all pregnant women with a record of syphilis in the clinical history and/or positive VDRL test during prenatal care or during hospitalization for

delivery, with confirmation through a rapid test. The identification of the cases occurred at the Regional Hospital of the Southwest Walter Alberto Pecóits, in Paraná, during hospitalization, through daily contact with the nurse responsible for the maternity ward, and through tracking in the hospitalization records, medical records, and prenatal cards of the puerperal women.

For the controls, pregnant and postpartum women with no history of syphilis in the anamnesis and with non-reactive results in the VDRL diagnostic tests and rapid confirmatory test were selected. The controls were recruited at the same time of data collection of cases, at the same establishment and were matched by age.

## Population and sample

The estimated population residing in the municipalities covered by the 8th Local Health Authority was 337,703 inhabitants in 2018 [10]. Women of childbearing age, defined as having a lower limit between 10 and 15 years and an upper limit between 44 and 49 years [11], accounted for 62.7% (106,145) of the entire female population (169,262). According to official indicators from the Ministry of Health regarding syphilis in Brazilian municipalities [12], from 2018, 102 cases of pregnant women with syphilis were reported, representing an incidence rate of 9.6% (if calculated based on the population of childbearing age) and 3.0% (if calculated based on the general population). In this study, the sample size estimate was performed using the *Stalcalc* program of the EpiInfo statistical package, version 6.04, based on a previous study [13], with a confidence level of 95% and power of 80%, as well as an estimated odds ratio (OR) of approximately 2.7 in relation to sociodemographic, behavioral, clinical, obstetric, prenatal, and maternity healthcare factors for syphilis in women. A ratio of one case to two controls was chosen, and the possibility of 10% loss or refusal was considered. Based on these criteria, it was estimated that 279 pregnant women would be required, including 93 cases and 186 controls. For each case, two age-matched controls were selected, considering a 5-year younger or older margin and the same period of admission to the maternity ward.

## Criteria selection

The classification between cases and controls was established after performing the Venereal Disease Research Laboratory (VDRL) non-treponemal screening test, which was confirmed with a rapid test (Bioclin-QUIBASA RT Syphilis). According to the Technical Manual for the Diagnosis of Syphilis [12], syphilis in pregnant women can be diagnosed through rapid or conventional treponemal tests [enzyme-linked immunosorbent assay (ELISA), fluorescent treponemal antibody test absorption test (FTA-Abs), or Treponema pallidum hemagglutination (TPHA)] and non-treponemal tests [VDRL, rapid plasma reagin (RPR), or Toluidine Red Unheated Serum Test (TRUST). All pregnant women with positive VDRL results and reactive rapid rests were included in the study, regardless of the clinical stage of syphilis (latent, primary, secondary, and tertiary).

## Study variables, instruments, and data collection procedures

The dependent variable was the syphilis infection status. The independent variables were categorized as follows: (a) sociodemographic data (age at the time of the interview, marital status, race/color, per capita family income based on the Brazilian minimum wage (MW), educational attainment, job remuneration, and insertion in the labor market); (b) behavioral (age at first sexual intercourse and first pregnancy, number of previous pregnancies and sexual partners, frequency of condom use, age at which smoking started, use of alcohol or illicit drugs, and use of illicit drugs by current partner); and (c) clinical, obstetric, and prenatal and maternity health care history (i.e., history of abortion, previous sexually transmitted infection, number of

prenatal consultations, and gestational age). The medical records were examined to confirm syphilis infection.

To assist the interview, a set of questionnaires developed and validated by Macedo [13] was used for data collection. The questionnaire was based on national and international research and validated in the Brazilian context. The researchers collected data through interviews in a private room at the maternity hospital, organized through daily contact with a nurse in the ward. For each sample, it took an average of 40 min to complete. Secondary data were collected from the admission records, medical records, and prenatal registration cards of postpartum women. When it was impossible to interview the women at the time of hospitalization, the interviews were conducted during childcare consultations at a specialty center for children exposed to syphilis.

To perform the non-treponemic test-VDRL, a serum sample was collected by the Biolabor Clinical Analysis Laboratory, which provides services to the hospital. The test was performed according to the manufacturer's instructions (Wiener Lab, Rosario, Argentina). For the rapid treponemal test, a digital puncture was performed to collect whole blood samples using the Bio Syphilis Kit (Bioclin, Belo Horizonte, Brazil) according to the manufacturer's instructions. All tests were performed as part of routine laboratory tests at Walter Alberto Pecóits Southeast Regional Hospital. The tests were collected from all pregnant women admitted for childbirth, regardless of gestational age, and VDRL was performed by the laboratory and rapid tests by trained professionals for both cases and controls.

## Statistical analysis

Descriptive statistical analyses were performed, including frequency (%), mean, amplitude, median, standard deviation, and confidence intervals. Bivariate analyses, such as Fisher's exact test and Pearson's chi-squared test, were used to identify factors associated with gestational syphilis. Those with a value of $p < 0.20$ were included in multivariate binary logistic regression to define risk factors for gestational syphilis, with $p < 0.05$. considered significant. The same strategy as in a previous study [13] was used to introduce the variables. A block modeling process, where the first block consisted of sociodemographic variables, the second block introduced behavioral variables, and the third block variables related to healthcare in pregnancy and maternity, was applied, considering possible risk factors for gestational syphilis. All data were analyzed using SPSS version 25.0.

## Ethical considerations

This study was approved by the Research Ethics Committee of the National Research Ethics Commission (CONEP) under number 4.242.982. All precautions were taken to ensure confidentiality and the secrecy of information. Before each interview, written consent was obtained from the participants after they read the free and informed consent forms.

## Results

Of the 291 potentially eligible women identified at the participating maternity hospitals, 12 (4.12%) were excluded from the study because of divergent results, refusal, and missing information that prevented their participation. Therefore, 93 patients and 186 controls were included in this study. The average age of the pregnant women was 25.70 years ± 6.4 years. On average, they had contact with alcohol at 15.53 years ± 3.13 years, cigarettes at 16.34 years ± 3.42 years, first sexual intercourse at 15.44 years ±2.19 years and became pregnant for the first time at 19.83 years ±4.76 years. The average income of pregnant women was USD

**Table 1. General characteristics of cases and controls and factors associated with syphilis in pregnant women admitted to the Southeast Regional Hospital, Paraná (n = 279).**

| Variables | Case (n = 93) | | Control (n = 186) | | p-value |
|---|---|---|---|---|---|
| | N | % | n | % | |
| **Sociodemographic variables** | | | | | |
| **Age group** | | | | | 0.654 |
| 14 to 30 years old | 73 | 78.5 | 140 | 75.3 | |
| 31 to 45 years old | 20 | 21.5 | 46 | 24.7 | |
| **Marital status** | | | | | 0.012 |
| Does not live with a partner | 22 | 23.9 | 21 | 11.4 | |
| Lives with a partner | 70 | 76.1 | 163 | 88.6 | |
| **Race/skin color** | | | | | 0.003 |
| White | 42 | 45.2 | 120 | 64.5 | |
| Other | 51 | 54.8 | 66 | 35.5 | |
| **Income** | | | | | 0.208 |
| Up to R$2.000 | 48 | 51.6 | 79 | 42.5 | |
| More than R$2,000 | 45 | 48.4 | 107 | 57.5 | |
| **Educational level of pregnant women** | | | | | 0.285 |
| Incomplete High School | 49 | 52.7 | 83 | 44.6 | |
| Complete High School | 36 | 38.7 | 77 | 41.4 | |
| Complete Higher Education | 8 | 8.6 | 26 | 14.0 | |
| **Education of the head of the family** | | | | | 0.304 |
| Incomplete High School | 50 | 58.1 | 88 | 48.1 | |
| Complete High School | 29 | 33.7 | 77 | 42.1 | |
| Complete Higher Education | 7 | 8.1 | 18 | 9.8 | |
| **Currently working** | | | | | 0.138 |
| Yes | 38 | 40.9 | 95 | 51.1 | |
| No | 55 | 59.1 | 91 | 48.9 | |
| **Behavioral variables** | | | | | |
| **Religion** | | | | | 0.320 |
| Catholic | 61 | 65.6 | 128 | 68.8 | |
| Evangelical | 23 | 24.7 | 49 | 26.3 | |
| Other | 9 | 9,7 | 9 | 4.8 | |
| **Age of first sexual intercourse** | | | | | 0.364 |
| Younger than 14 years | 14 | 15.4 | 20 | 10.8 | |
| 14 years old or older | 77 | 84.6 | 166 | 89.2 | |
| **Age at first pregnancy** | | | | | 0.042 |
| Younger than 16 years | 21 | 22.6 | 23 | 12.4 | |
| 16 years old or older | 72 | 77.4 | 163 | 87.6 | |
| **Number of sexual partners in the previous year** | | | | | <0.001 |
| Only one | 68 | 73.1 | 171 | 91.9 | |
| More than one | 25 | 26.9 | 15 | 8.1 | |
| **Condom use** | | | | | 0.087 |
| Yes | 41 | 44.1 | 61 | 32.8 | |
| No | 52 | 55.9 | 125 | 67.2 | |
| **Smoking** | | | | | <0.001 |
| No | 45 | 48.4 | 144 | 77.4 | |
| No, but has previously smoked | 33 | 35.5 | 34 | 18.3 | |
| Yes | 15 | 16.1 | 8 | 4.3 | |

*(Continued)*

**Table 1.** (Continued)

| Variables | Case (n = 93) | | Control (n = 186) | | p-value |
|---|---|---|---|---|---|
| | N | % | n | % | |
| **Alcohol consumption** | | | | | 0.026 |
| No | 80 | 86.0 | 176 | 94.6 | |
| Yes | 13 | 14.0 | 10 | 5.4 | |
| **Drug usage** | | | | | 0.001 |
| No | 81 | 87.1 | 180 | 97.8 | |
| Yes | 12 | 12.9 | 4 | 2.2 | |
| **Partner's drug usage** | | | | | 0.049 |
| No | 84 | 90.3 | 180 | 96.8 | |
| Yes | 9 | 9.7 | 6 | 3.2 | |
| **Healthcare during prenatal and maternity** | | | | | |
| **Number of prenatal consultations** | | | | | 0.863 |
| 0 to 6 | 16 | 17.2 | 29 | 15.6 | |
| 7 or more | 77 | 82.8 | 157 | 84.4 | |
| **Starting prenatal care** | | | | | 0.323 |
| Until the 1[st] trimester | 75 | 80.6 | 160 | 86.0 | |
| After the 1[st] trimester | 18 | 19.4 | 26 | 14.0 | |
| **Prenatal location** | | | | | 1.000 |
| Family and Healthcare Program | 85 | 91.4 | 170 | 91.4 | |
| Other | 8 | 8.6 | 16 | 8.6 | |
| **Received prenatal registration card at the 1[st] consultation** | | | | | 0.470 |
| Yes | 84 | 90.3 | 174 | 93.5 | |
| No | 9 | 9.7 | 12 | 6.5 | |
| **History of abortions** | | | | | 0.654 |
| No | 82 | 88.2 | 158 | 85.4 | |
| Yes | 11 | 11.8 | 27 | 14.6 | |
| **History of STI[a]** | | | | | <0.001 |
| No | 67 | 72.8 | 179 | 96.8 | |
| Yes | 25 | 27.2 | 6 | 3.2 | |

[a]STI: sexually transmitted infection.

490.52 ± 357.94. The variable age at first pregnancy exhibited a statistically significant difference between the cases and controls.

Regarding sociodemographic variables, we observed a higher frequency of cases among pregnant women who did not live with a partner and those who had a race/skin color other than white. Regarding behavioral variables, the cases showed a higher frequency of pregnancy before the age of 16 years, a higher number of sexual partners, higher consumption of illicit drugs, and higher consumption of illicit drugs by a sexual partner. Only a history of STIs was associated with variables related to health care in prenatal and maternity periods, and was also more frequent among cases compared to controls, as shown in Table 1.

Following bivariate analyses, Table 2 shows the crude and adjusted binary logistic regression models. Of all the variables introduced into the model, only four remained significant and independently associated with syphilis. Non-white women were twice as likely to have such outcomes (OR: 2.12; 95%CI: 1.19–3.80). In addition, women who had more than one sexual

**Table 2. Crude and adjusted models of factors associated with syphilis in pregnant women admitted to the Southeast Regional Hospital, Paraná (n = 279).**

| Risk factors | OR$_{crude}$ (95%CI) | p-value | OR$_{adjusted}$ (95%CI) | p-value |
|---|---|---|---|---|
| **Sociodemographic variables** | | | | |
| **Marital status** | | | | |
| Does not live with a partner | 2.44 (1.26–4.72) | 0.008 | ns | ns |
| Lives with a partner | 1 | | | |
| Race/skin color | | | | |
| White | 1 | | 1 | |
| Other | 2.21 (1.33–3.67) | 0.002 | 2.12 (1.19–3.80) | 0.011 |
| **Currently working** | | | | |
| Yes | 1 | | | |
| No | 1.51 (0.91–2.50) | 0.108 | ns | ns |
| **Behavioral variables** | | | | |
| **Age at first pregnancy** | | | | |
| Younger than 16 years | 2.07 (1.08–3.97) | 0.029 | ns | ns |
| 16 years old or older | 1 | | | |
| **Number of sexual partners in the previous year** | | | | |
| Only 1 | 1 | | 1 | |
| More than 1 | 4.19 (2.08–8.43) | <0.001 | 3.69 (1.70–8.00) | 0.001 |
| **Condom use** | | | | |
| Yes | 1 | | | |
| No | 0.62 (0.37–1.03) | 0.066 | — | — |
| **Smoking** | | | | |
| No | 1 | | 1 | |
| No, but has previously smoked | 3.11 (1.73–5.57) | <0,001 | 2.07 (1.07–4.01) | 0.030 |
| Yes | 6.00 (2.39–15.07) | <0.001 | 4.31 (1.55–11.98) | 0.005 |
| **Alcohol consumption** | | | | |
| No | 1 | | | |
| Yes | 2.86 (1.20–6.80) | 0.017 | ns | ns |
| **Drug usage** | | | | |
| No | 1 | | | |
| Yes | 6.67 (2.09–21.3) | 0.001 | Ns | ns |
| **Partner's drug usage** | | | | |
| No | 1 | | | |
| Yes | 3.21 (1.11–9.32) | 0.032 | ns | ns |
| **Health care during prenatal and maternity periods** | | | | |
| **History of STI[a]** | | | | |
| No | 1 | | | |
| Yes | 11.13 (4.37–28.3) | <0.001 | 10.87 (4.04–29.27) | <0.001 |

[a]STI: sexually transmitted infection. ns–non-significant.

partner in the previous year had higher odds of gestational syphilis than those who had only one partner (OR: 3.69; 95%CI: 1.70–8.00). The odds of having gestational syphilis were 4.31 times (95%CI: 1.55–11.98) if a woman reported current smoking habits compared with those who did not smoke. The same pattern was found in those who reported smoking in the past (OR: 2.07; 95%CI: 1.07–4.01). Finally, a history of STI increased the possibility of gestational syphilis (OR: 10.87; 95%CI: 4.04–29.27) when compared to participants without previous STI infection.

## Discussion

Syphilis is an infection that has highly cost-effective screening and treatment capable of preventing adverse outcomes associated with infection in pregnancy [12]. In this sense, recognizing the sociodemographic, behavioral, and care factors associated with gestational syphilis allows for early diagnosis and management, thereby avoiding unwanted gestational outcomes.

The implementation of broad public policies that promote the improvement of living conditions to address the social determinants that contribute to the prevention of STIs, including counseling for risk reduction, increased access to condoms, and early treatment, can help reduce the incidence rate of syphilis among pregnant women or those aiming pregnancy [14, 15]. On the other hand, the development of intersectoral actions aimed at syphilis prevention and health education should be present in the various activities of health professionals to provide the exchange of information with users about the disease and its implications, especially during pregnancy [4].

In this study, we observed a higher frequency of sociodemographic (absence of a partner and non-white race) and behavioral risk factors (use of licit/illicit drugs and pregnancy before the age of 16 years) among the cases. Trivedi et al. [4] stated that these factors interfere with adherence to prenatal care and–consequently–affect syphilis screening and treatment, as well as contribute to loss of follow-up and reinfection after treatment. Our findings are consistent with the study conducted by Benzaken et al. (2020) in Brazilian state capitals, in which an association was observed between age under 20 years, non-white race, and marital status without a partner with inadequate prenatal care [16]. This is possibly due to the fact that women with low education and low socioeconomic status lack health care, especially preventive measures– which hinders the interruption of the chain of transmission [17, 18]. This reinforces the fact that patient education is essential to increase the dissemination of information about syphilis infection and the possibility of treating this infection [4, 15].

In our sample, the history of STIs increased the chance of syphilis by more than 10 times. For this reason, in the literature, it is recommended that women with positive results for syphilis undergo detailed anamnesis and physical examination, as well as tests for other STIs– including HIV [19]. Likewise, we consider it essential to reinforce the guidance on the use of condoms in risky sexual relations for pregnant women–especially if they have contact with more than one partner.

We noticed that partner-related behavioral risk variables (number of sexual partners and drug use) were also more frequent among the cases. A study conducted by Lima et al. (2009) found similar findings, as well as related the higher prevalence of gestational syphilis in women with partners with alcohol abuse and STIs [20]. This is probably due to the lack of standardized treatment for sexual partners, which contributes to exposing pregnant women to recurrent or persistent infections [21]. Similarly, Campos et al. (2012) noted that few partners provide adequate treatment, due to insufficient adherence to accompany women to prenatal consultations and inadequate reception in health services [22].

Being a smoker and an ex-smoker increased the chance of having syphilis by 331% and 107%, respectively. Other studies have also found a higher prevalence of tobacco use among pregnant women with syphilis [23, 24] Smoking is a high-risk marker for syphilis infection, especially when associated with increased sexual activity, greater number of sexual partners throughout life, lower use of condoms, and earlier age at first sexual intercourse and pregnancy [25, 26].

Our study has some limitations. First, this study was conducted in a tertiary health care service, a fact that may contribute to the high prevalence of gestational syphilis due to the higher frequency of referral of these patients to specialized services. In addition, our data did not

include information on pregnancy outcome. Thus, it was not possible to statistically associate the factors associated with prenatal care with congenital syphilis or adverse pregnancy outcomes. Also, the instrument used to investigate risk factors and characterize the group's profile may have resulted in a bias of malfeasance, especially in the behavioral variables, considering that the answers could contain inaccuracies or omissions for various reasons, such as shame, fear, or forgetfulness. Moreover, the research was conducted in a public hospital unit, which may limit the extrapolation of our data to other settings. Another relevant point to be considered is that other pre-existing medical conditions may be linked to gestational syphilis, which can lead to confusion, especially when analyzing variables related to prenatal care and maternity.

## Conclusion

We identified several risk factors (sociodemographic, behavioral, and healthcare-related) that should be considered in the management and formulation of health care policies aimed at the prevention, screening, and treatment of sexually transmitted infections to reduce adverse outcomes associated with pregnancy, particularly congenital syphilis. The recognition of these risk factors (race/ethnicity, number of sexual partners, current and past tobacco use, and previous STI) by health professionals can enable the implementation of educational activities on sexuality, aiming to instruct the population at risk about the use of condoms, the importance of early detection, and appropriate treatment during pregnancy. Interventions and approaches that emphasize human rights, gender equality, and overcoming barriers can promote equitable access to health services for diverse populations and in different contexts, integrating the prevention and management of gestational syphilis with sexual and reproductive health.

## Supporting information

**S1 Data.** https://doi.org/10.6084/m9.figshare.25966969.v1.
(SAV)

## Acknowledgments

We would like to acknowledge the Hospital Regional do Sudoeste Walter Alberto Pecóits (Walter Alberto Pecóits Southeastern Regional Hospital), in the state of Paraná, Brazil, for allowing us to conduct this study within their premises.

## Author Contributions

**Conceptualization:** Fernando Braz Pauli, Valdir Spada Júnior, Renan William Mesquita, Guilherme Welter Wendt, Paulo Cezar Nunes Fortes, Harapan Harapan, Lirane Elize Defante Ferreto.

**Data curation:** Fernando Braz Pauli, Valdir Spada Júnior, Renan William Mesquita, Guilherme Welter Wendt, Paulo Cezar Nunes Fortes, Harapan Harapan, Lirane Elize Defante Ferreto.

**Formal analysis:** Fernando Braz Pauli, Valdir Spada Júnior, Renan William Mesquita, Guilherme Welter Wendt, Paulo Cezar Nunes Fortes, Harapan Harapan, Lirane Elize Defante Ferreto.

**Funding acquisition:** Fernando Braz Pauli, Valdir Spada Júnior, Renan William Mesquita, Guilherme Welter Wendt, Paulo Cezar Nunes Fortes, Harapan Harapan, Lirane Elize Defante Ferreto.

**Investigation:** Fernando Braz Pauli, Valdir Spada Júnior, Renan William Mesquita, Guilherme Welter Wendt, Paulo Cezar Nunes Fortes, Harapan Harapan, Lirane Elize Defante Ferreto.

**Methodology:** Fernando Braz Pauli, Valdir Spada Júnior, Renan William Mesquita, Guilherme Welter Wendt, Paulo Cezar Nunes Fortes, Harapan Harapan, Lirane Elize Defante Ferreto.

**Project administration:** Fernando Braz Pauli, Valdir Spada Júnior, Renan William Mesquita, Guilherme Welter Wendt, Paulo Cezar Nunes Fortes, Harapan Harapan, Lirane Elize Defante Ferreto.

**Resources:** Fernando Braz Pauli, Valdir Spada Júnior, Renan William Mesquita, Guilherme Welter Wendt, Paulo Cezar Nunes Fortes, Harapan Harapan, Lirane Elize Defante Ferreto.

**Software:** Fernando Braz Pauli, Valdir Spada Júnior, Renan William Mesquita, Guilherme Welter Wendt, Paulo Cezar Nunes Fortes, Harapan Harapan, Lirane Elize Defante Ferreto.

**Supervision:** Fernando Braz Pauli, Valdir Spada Júnior, Renan William Mesquita, Guilherme Welter Wendt, Paulo Cezar Nunes Fortes, Harapan Harapan, Lirane Elize Defante Ferreto.

**Validation:** Fernando Braz Pauli, Valdir Spada Júnior, Renan William Mesquita, Guilherme Welter Wendt, Paulo Cezar Nunes Fortes, Harapan Harapan, Lirane Elize Defante Ferreto.

**Visualization:** Fernando Braz Pauli, Valdir Spada Júnior, Renan William Mesquita, Guilherme Welter Wendt, Paulo Cezar Nunes Fortes, Harapan Harapan, Lirane Elize Defante Ferreto.

**Writing – original draft:** Fernando Braz Pauli, Valdir Spada Júnior, Renan William Mesquita, Guilherme Welter Wendt, Paulo Cezar Nunes Fortes, Harapan Harapan, Lirane Elize Defante Ferreto.

**Writing – review & editing:** Fernando Braz Pauli, Valdir Spada Júnior, Renan William Mesquita, Guilherme Welter Wendt, Paulo Cezar Nunes Fortes, Harapan Harapan, Lirane Elize Defante Ferreto.

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
