## [Decision Letter · Decision Letter 0]

19 Apr 2024

PONE-D-23-39335GESTATIONAL SYPHILIS IN A TERTIARY HEALTH SERVICE IN PARANÁ, BRAZIL: A CASE-CONTROL STUDYPLOS ONE

Dear Dr. Defante Ferreto,

Thank you for submitting your manuscript to PLOS ONE. After careful consideration, we feel that it has merit but does not fully meet PLOS ONE’s publication criteria as it currently stands. Therefore, we invite you to submit a revised version of the manuscript that addresses the points raised during the review process.

We look forward to receiving your revised manuscript.

Kind regards,

Thales Philipe Rodrigues da Silva, Ph.D

Academic Editor

PLOS ONE

2. Please amend your authorship list in your manuscript file to include author Lirane Elize Defante Ferreto, Fernando Braz Pauli, Valdir Spada Júnior, Renan William Mesquita, Guilherme Welter Wendt, Paulo Cezar Nunes Fortes and Harapan Harapan. 

4. Please remove your figures from within your manuscript file, leaving only the individual TIFF/EPS image files, uploaded separately. These will be automatically included in the reviewers’ PDF.

5. Please include your tables as part of your main manuscript and remove the individual files. Please note that supplementary tables (should remain/ be uploaded) as separate ""supporting information"" files.

Additional Editor Comments:

Dear Editor.

Thank you for the opportunity to evaluate the article in question.

Thank you in advance.

In view of the reviewers' assessments, I am sending the article for further revision by the authors and future resubmission.

Sincerely,

Reviewers' comments:

Reviewer's Responses to Questions

**Comments to the Author**

1. Is the manuscript technically sound, and do the data support the conclusions?

Reviewer #1: Yes

Reviewer #2: Partly

2. Has the statistical analysis been performed appropriately and rigorously? 

Reviewer #1: Yes

Reviewer #2: Yes

3. Have the authors made all data underlying the findings in their manuscript fully available?

Reviewer #1: Yes

Reviewer #2: No

4. Is the manuscript presented in an intelligible fashion and written in standard English?

Reviewer #1: Yes

Reviewer #2: Yes

5. Review Comments to the Author

Reviewer #1: Thank you for the opportunity to review your work which focuses on identifying factors associated with syphilis in pregnant women. It is an ambispective paired case-control study conducted over a year, comparing syphilis-positive pregnant women with matched controls. The study uses interviews and medical record reviews, analyzing the data with binary logistic regression.

General Comments:

1. Significance: The research addresses an important public health issue, offering valuable insights into gestational syphilis, which is crucial for developing effective healthcare policies.

2. Methodology: The case-control study design is appropriate for the research objective. However, the methodology could be more detailed regarding participant selection and data collection processes.

3. Analysis and Interpretation: The use of binary logistic regression for data analysis is suitable. The interpretation of results appears to be consistent with the data presented.

Specific Comments:

1. Introduction: The introduction effectively sets the context for the study, but it might benefit from a more detailed discussion of previous research on gestational syphilis in Brazil.

2. Methods: Adequately detailed, but could provide more information on the sampling process and questionnaire design.

3. Discussion: This section effectively relates the findings to the wider context of syphilis in pregnancy, but could delve deeper into implications for clinical practice and policy.

4. Conclusion: Summarizes the main findings and implications of the study well, though it could more explicitly state the limitations of the study. good job.

Reviewer #2: This is a useful study that examined the social background and identified risk factors to prevent the increasing syphilis infections.

There are a few things I would like to confirm.

P10 & P11

Which reference does the term "Macedo" refer to? I wanted to check, but I couldn't find it.

“To assist the interview, a set of questionnaires developed and validated by

Macedo[10] was used for data collection.”

“The same strategy as in a previous study (Macedo, 2015) was used to introduce the variables. “

When was the questionnaire conducted? If it is a standardized questionnaire, please attach the English version as a reference.

The principle of a confirmatory test for syphilis involves either ELISA, FTA-ABS, or TPHA？

If syphilis screening is being conducted, please specify the week of gestation. Also, have all pregnant women in the control group been confirmed as negative for syphilis?

Can all stages of syphilis be interpreted as either primary or secondary syphilis?

The discussion covers both the prevention of infection during pregnancy and the prevention of infection before pregnancy. What is the main focus of the study—preventing infection before pregnancy or during pregnancy? If syphilis is diagnosed early in pregnancy, the primary challenge may be to consider preventative measures before pregnancy. What are authors thoughts on this?

The discussion addresses inadequate prenatal care and precautions during pregnancy; however, if the study aims to examine the background factors of pregnant women who were infected with syphilis before pregnancy, the discussion may not be a reflection of the study results.

In the conclusion, clearly and concisely present the risk factors identified in this study. Terms such as congenital syphilis or treatment may not be directly drawn from these results.

There is inconsistency in the citation style of references within the main text. Please ensure they are uniform.

The reference list contains entries in Portuguese, so please standardize the language to English.

6. PLOS authors have the option to publish the peer review history of their article (what does this mean?). If published, this will include your full peer review and any attached files.

Reviewer #1: No

Reviewer #2: No

---

## [Author Response · Author response to Decision Letter 0]

18 May 2024

RESPONSE LETTER

To the editorial board:

Many thanks for considering the manuscript “PONE-D-23-39335” for publication. On behalf of all the collaborators, please accept our gratitude in receiving such constructive feedback on our paper. We broaden this sentiment to the reviewers, who gave us not only their precious time but also their insightful analysis. 

In summary, all the requests were addressed. Nonetheless, this file contains the responses to every comment and suggestions, as detailed below. 

Changes to the revised manuscript are highlighted with the aid of ‘Track Changes MS Word function’. Following the requirements, two separate files are attached to the system, namely: ‘Revised Manuscript with Track Changes’ and ‘Manuscript’ with the inclusion of the title page within these files as requested. 

Please, do not hesitate in contacting us if there are any other issues involving our article that should be improved.

Yours sincerely,

Lirane E F Ferreto, PhD

 EDITORIAL OFFICE COMMENTS

Comment e.1: Please ensure that your manuscript meets PLOS ONE's style requirements, including those for file naming. The PLOS ONE style templates can be found at https://journals.plos.org/plosone/s/file?id=wjVg/PLOSOne_formatting_sample_main_body.pdf and

Response e.1: The manuscript has been reviewed and updated to adhere to PLOS ONE’s style requirements. We carefully investigated the template. All the formatting requirements seems to be correct now – including file naming, Tables and Figures.

Comment e.2: Please amend your authorship list in your manuscript file to include author Lirane Elize Defante Ferreto, Fernando Braz Pauli, Valdir Spada Júnior, Renan William Mesquita, Guilherme Welter Wendt, Paulo Cezar Nunes Fortes and Harapan Harapan. 

Response e.2: We apologize for any inconvenience this might have caused. We revised all the author details and updated the Title Page, as well as the information within the Editorial Manager submission system to properly reflect the authors’ information.

Comment e.3: Please amend your list of authors on the manuscript to ensure that each author is linked to an affiliation. Authors’ affiliations should reflect the institution where the work was done (if authors moved subsequently, you can also list the new affiliation stating “current affiliation:….” as necessary).

Response e.3: This error has been corrected. 

Comment e.4: Please remove your figures from within your manuscript file, leaving only the individual TIFF/EPS image files, uploaded separately. These will be automatically included in the reviewers’ PDF.

Response e.4: Once again, apologies for this mistake. Corrective measures have been taken to address this issue. Precisely, figure 1 has been removed as it might contain copyright-sensitive information and, overall, does not add substantially to the manuscript.

Comment e.5: Please include your tables as part of your main manuscript and remove the individual files. Please note that supplementary tables (should remain/ be uploaded) as separate ""supporting information"" files.

Response e.5: Apologies for this mistake. Corrective measures have been taken to address this issue. 

Comment e.6: We note your current Data Availability statement is: "The data underlying the results presented in this study are available from the Western Paraná State University Research Ethics Committee based on individual requests since sensitive information protected by law (medical records) were recorded.

Researchers who wish to access confidential data are encouraged to contact Dr. Lirane E. D. Ferreto (Director, Health Sciences Center, Western Paraná State University; lirane.ferreto@unioeste.br) to discuss how their data request can be facilitated.;

Tick here if the URLs/accession numbers/DOIs will be available only after acceptance of the manuscript for publication so that we can ensure their inclusion before publication."

Response e.6: Apologies for this mistake. Data will be made public after acceptance. 

 REVIEWER 1

Comments from Reviewer #1: Thank you for the opportunity to review your work which focuses on identifying factors associated with syphilis in pregnant women. It is an ambispective paired case-control study conducted over a year, comparing syphilis-positive pregnant women with matched controls. The study uses interviews and medical record reviews, analyzing the data with binary logistic regression.

General Comments: 1. Significance: The research addresses an important public health issue, offering valuable insights into gestational syphilis, which is crucial for developing effective healthcare policies. 2. Methodology: The case-control study design is appropriate for the research objective. However, the methodology could be more detailed regarding participant selection and data collection processes. 3. Analysis and Interpretation: The use of binary logistic regression for data analysis is suitable. The interpretation of results appears to be consistent with the data presented.

Specific Comments: 1. Introduction: The introduction effectively sets the context for the study, but it might benefit from a more detailed discussion of previous research on gestational syphilis in Brazil. 2. Methods: Adequately detailed, but could provide more information on the sampling process and questionnaire design. 3. Discussion: This section effectively relates the findings to the wider context of syphilis in pregnancy, but could delve deeper into implications for clinical practice and policy. 4. Conclusion: Summarizes the main findings and implications of the study well, though it could more explicitly state the limitations of the study. good job.

Responses to Reviewer #1: We thank the reviewer for these comments. We addressed all of them and a brief overview of the changes performed are detailed next. Firstly, the reviewer suggested to update the introduction as to include a “more detailed discussion of previous research on gestational syphilis in Brazil”. In this regard, we inform that we included two paragraphs in the introduction (lines 59-69), that are clearly marked in red color.

Next, we were asked to provide more information on the sampling process and questionnaire design (Reviewer #1, Specific comment 2). In the revised manuscript, we improved the methods section as whole, also because the second reviewer requested further information about these aspects. We hope that all the information added to the text is considered to be sufficient. However, if further adjustments are deemed necessary, please, just let us know.

The specific comments 3 and 4 requested the necessity of adding a more in-depth discussion in terms of clinical practice and policy, as well as being more explicit when stating our limitations. In this regard, two new paragraphs were included in the discussion (lines 231-239), as well as a complete revision of the limitations of the study (lines 275-287). If further adjustments are deemed necessary, please, just let us know.

 REVIEWER 2

Comments from Reviewer #2: This is a useful study that examined the social background and identified risk factors to prevent the increasing syphilis infections.

There are a few things I would like to confirm.

1) P10 & P11. Which reference does the term "Macedo" refer to? I wanted to check, but I couldn't find it. “To assist the interview, a set of questionnaires developed and validated by Macedo[10] wa used for data collection.” “The same strategy as in a previous study (Macedo, 2015) was used to introduce the variables”.

2) When was the questionnaire conducted? If it is a standardized questionnaire, please attach the English version as a reference.

3) The principle of a confirmatory test for syphilis involves either ELISA, FTA-ABS, or TPHA？

4) If syphilis screening is being conducted, please specify the week of gestation. Also, have all pregnant women in the control group been confirmed as negative for syphilis?

5) Can all stages of syphilis be interpreted as either primary or secondary syphilis?

6) The discussion covers both the prevention of infection during pregnancy and the prevention of infection before pregnancy. What is the main focus of the study—preventing infection before pregnancy or during pregnancy? If syphilis is diagnosed early in pregnancy, the primary challenge may be to consider preventative measures before pregnancy. What are authors thoughts on this?

7) The discussion addresses inadequate prenatal care and precautions during pregnancy; however, if the study aims to examine the background factors of pregnant women who were infected with syphilis before pregnancy, the discussion may not be a reflection of the study results.

8) In the conclusion, clearly and concisely present the risk factors identified in this study. Terms such as congenital syphilis or treatment may not be directly drawn from these results.

9) There is inconsistency in the citation style of references within the main text. Please ensure they are uniform. The reference list contains entries in Portuguese, so please standardize the language to English.

Responses to Reviewer #2: First of all, we would like to thank the reviewer for these suggestions and overall assessment of the manuscript. The responses to the 9 points raised by the Reviewer #2 are detailed next.

Points 1 and 2 - In respect to the questionnaire used, we indeed failed to provide more detailed information about it, and the reference number did not match the original study. We amended this section, and all the changes are clearly marked in red. The questionnaire developed by Macedo is not available in English, at least yet. We did not hear from the author regarding any ongoing work regarding the translation of the material to English. We also believe that such an effort would be astronomical, since the interview is extremely comprehensive and was the main product of her PhD work (https://repositorio.ufpe.br/bitstream/123456789/16160/1/Doutorado_POSCA_VilmaMacedo_2015.pdf). Nonetheless, if the reviewer wants a draft of the 204 questions of the instrument (pages 114-125 from this document https://repositorio.ufpe.br/bitstream/123456789/16160/1/Doutorado_POSCA_VilmaMacedo_2015.pdf), in English, please, let us know. We would need, however, permission from the original author to do so.

Points 3, 4 and 5 – A revision of the methods section was performed as a whole, and all these points were addressed. Particularly, the information included in lines 128-134 address the questions raised by the Reviewer #2. If further adjustments are deemed necessary, please, just let us know.

Points 6 and 7 – The questions as to whether the study aimed to contribute with information for syphilis in pregnant women or for preventing infections despite pregnancy status are indeed quite important. In our view, the results can be useful insightful both ways. To account for this, we included these two perspectives in the discussion (lines 231-238).

Points 8 and 9 – The citation style has been amended, and your suggestions for a more direct conclusion has been considered. Therefore, the conclusion directly recaps the risk factors identified in the study. Many thanks for your assessment and if you still think that some parts of the text should be improved, please, just let us know.

- - - - 

Yours sincerely,

The authors

---

## [Decision Letter · Decision Letter 1]

3 Jun 2024

Gestational syphilis in a tertiary health service in Paraná, Brazil: a case-control study

PONE-D-23-39335R1

Dear Dr. Defante Ferreto,

We’re pleased to inform you that your manuscript has been judged scientifically suitable for publication and will be formally accepted for publication once it meets all outstanding technical requirements.

Kind regards,

Thales Philipe Rodrigues da Silva, Ph.D

Academic Editor

PLOS ONE

Additional Editor Comments (optional):

Dear editor,

I am pleased to inform you that manuscript has been accepted for publication.  

My comments, and any additional reviewer comments, can be found below.

Thanks to the authors for addressing all of the reviewer comments, I have accepted the paper for publication.

The only modification required will be to improve the quality of the image resolution.

Kind regards

Reviewers' comments:

Reviewer's Responses to Questions

**Comments to the Author**

1. If the authors have adequately addressed your comments raised in a previous round of review and you feel that this manuscript is now acceptable for publication, you may indicate that here to bypass the “Comments to the Author” section, enter your conflict of interest statement in the “Confidential to Editor” section, and submit your "Accept" recommendation.

Reviewer #1: All comments have been addressed

Reviewer #2: All comments have been addressed

2. Is the manuscript technically sound, and do the data support the conclusions?

Reviewer #1: Yes

Reviewer #2: Yes

3. Has the statistical analysis been performed appropriately and rigorously? 

Reviewer #1: Yes

Reviewer #2: Yes

4. Have the authors made all data underlying the findings in their manuscript fully available?

Reviewer #1: Yes

Reviewer #2: Yes

5. Is the manuscript presented in an intelligible fashion and written in standard English?

Reviewer #1: Yes

Reviewer #2: Yes

6. Review Comments to the Author

Reviewer #1: Thanks for attentively addressing the comments from reviewers. It reads better to me and I appreciate your conscientiousness. Looking forward to reading from you in the future

Reviewer #2: Thanks to the revisions made by the authors, the content has become much clearer. I am grateful for their revisions.

7. PLOS authors have the option to publish the peer review history of their article (what does this mean?). If published, this will include your full peer review and any attached files.

Reviewer #1: **Yes: **Udoka Okpalauwaekwe

Reviewer #2: No

---

## [Editor Report · Acceptance letter]

29 Jul 2024

PONE-D-23-39335R1 

PLOS ONE

Dear Dr. Defante Ferreto, 

I'm pleased to inform you that your manuscript has been deemed suitable for publication in PLOS ONE. Congratulations! Your manuscript is now being handed over to our production team.

Kind regards, 

on behalf of

Dr. Thales Philipe Rodrigues da Silva 

Academic Editor

PLOS ONE